# Taxonomy and Phylogeny of *Cystolepiota* (Agaricaceae, Agaricales): New Species, New Combinations and Notes on the *C. seminuda* Complex

**DOI:** 10.3390/jof9050537

**Published:** 2023-04-30

**Authors:** Hua Qu, Ulrike Damm, Ya-Jun Hou, Zai-Wei Ge

**Affiliations:** 1CAS Key Laboratory for Plant Diversity and Biogeography of East Asia, Kunming Institute of Botany, Chinese Academy of Sciences, Lanhei Road 132, Kunming 650201, China; quhua@mail.kib.ac.cn (H.Q.); houyajun@mail.kib.ac.cn (Y.-J.H.); 2Yunnan Key Laboratory for Fungal Diversity and Green Development, Kunming Institute of Botany, Chinese Academy of Sciences, Kunming 650201, China; 3University of the Chinese Academy of Sciences, Beijing 100049, China; 4Department of Botany, Senckenberg Museum of Natural History Görlitz, PF 300 154, 02806 Görlitz, Germany; ulrike.damm@senckenberg.de

**Keywords:** molecular phylogeny, new combinations, new species, species delimitation, systematics

## Abstract

Species of *Cystolepiota* are known as diminutive lepiotaceous fungi with a worldwide distribution. Previous studies revealed that *Cystolepiota* is not monophyletic and preliminary DNA sequence data from recent collections suggested that several new species exist. Based on multi-locus DNA sequence data (the nuc rDNA internal transcribed spacer region ITS1-5.8S-ITS2, ITS; the D1–D2 domains of nuc 28S rDNA, LSU; the most variable region of the second-largest subunit of RNA polymerase II, *rpb2* and a portion of the translation–elongation factor 1-α. *tef1*), *C.* sect. *Pulverolepiota* forms a distinct clade separating from *Cystolepiota*. Therefore, the genus *Pulverolepiota* was resurrected and two combinations, *P. oliveirae* and *P. petasiformis* were proposed. With the integration of morphological characteristics, multi-locus phylogeny, and information on geography and habitat, two new species, viz. *C. pseudoseminuda* and *C. pyramidosquamulosa*, are described and *C. seminuda* was revealed to be a species complex containing at least three species, viz. *C. seminuda*, *C. pseudoseminuda*, and *Melanophyllum eryei*. In addition, *C. seminuda* was re-circumscribed and neo-typified based on recent collections.

## 1. Introduction

The genus *Cystolepiota* Singer (Agaricaceae, Basidiomycota) was erected by Singer and Digilio (1951) to accommodate tiny and delicate mushroom species with epithelioid pileus-coverings and basidiospores that are neither amyloid nor dextrinoid [1]. Several species with dextrinoid basidiospores were subsequently placed or combined into this genus., e.g., *C. bucknallii* (Berk. and Broome) Singer and Clémençon, *C. icterina* F.H. Møller ex Knudsen and *C. microspora* (Sacc.) Singer and Clémençon [2,3]. Singer and Clémençon (1972) further divided the genus in two sections: *C*. sect. *Pseudoamyloideae* Singer & Clemençon, accommodating species with dextrinoid basidiospores, and *C.* sect. *Cystolepiota* Singer, containing the remaining species [3]. *Pulverolepiota* Bon, a genus proposed by Bon (1993) to accommodate *Lepiota pulverulenta* Huijsman (synonym of *C. petasiformis* (Murrill) Vellinga) and *L. roseolanata* Huijsman, was transferred to *Cystolepiota* as the third section, *C.* sect. *Pulverolepiota* (Bon) Vellinga, characterized by slow-staining dextrinoid spores, the absence of cystidia, and elongated and inflated pileus covering elements [4,5,6].

Based on the similarity of the pileus covering structures, species of *Lepiota* sect. *Echinatae* Fayod were regarded as *Cystolepiota* species by some mycologists [2,7,8,9] and treated as *Cystolepiota* subgen. *Echinoderma* Locq. ex Bon, while other mycologists suggested these species should remain in *Lepiota* (Pers.) Gray [10,11,12,13,14,15]. In contrast, Bon (1991) placed these species in the new genus *Echinoderma* (Locq. ex Bon) Bon [16]. A phylogeny inferred from sequences of the nuc rDNA internal transcribed spacer region ITS1-5.8S-ITS2 (ITS) and the D1–D2 domains of nuc 28S rDNA (LSU) showed the genus *Cystolepiota* to be closely related to *L.* sect. *Echinatae*, *Melanophyllum* Velen. and *Pulverolepiota* [17]. According to this phylogeny, species of *L.* sect. *Echinatae* formed two clades: a “large-spored” group that included *L. aspera* (Pers.) Quél., *L. hystrix* F.H. Møller & J.E. Lange, and *L. perplexa* Knudsen and a “small-spored” group that included *L. jacobi* Vellinga & Knudsen, *L. echinacea* J.E. Lange, etc. However, in a recent multi-locus sequence analysis [18], the species of original *L.* sect. *Echinatae*, those with large, subcylindrical spores and that include *L. aspera*, the type species of this section, represented a sister taxon to a clade formed by *Cystolepiota* and *Melanophyllum* and were treated as *Echinoderma* (hereafter referred to as *Echinoderma* sensu stricto), while the species with smaller, ellipsoid-to-oblong spores remained in *Lepiota* (hereafter referred as “*L*. sect. *Echinatae*”). A further study showed *Cystolepiota* and *Melanophyllum* constitute a clade sister to *Smithiomyces* Singer and that species of both *Echinoderma* sensu stricto and *L*. sect. *Echinatae* have more distant relationships with *Cystolepiota* and therefore should not be included in *Cystolepiota* [19].

*Cystolepiota seminuda* (Lasch) Bon is considered to be distributed worldwide [20,21,22]. However, the variability of size and color of basidiomata, as well as its spore morphology, has given rise to considerable controversy on its taxonomic treatment [21]. The basionym *Agaricus seminudus* Lasch was described based on macroscopic characteristics by Lasch in 1828 [23] and was later transferred to *Lepiota*, *Cystoderma* Fayod, *Mastocephalus* Battarra (nom. inval.), and finally to *Cystolepiota* [24,25,26,27]. Quélet treated *L. seminuda* (Lasch) P. Kumm. as a variety of *L. sistrata* (Fr.) Quél., a species originally described in Sweden as *A. sistratus* Fr. and later combined with *Cystolepiota* as well [28,29]. Several authors considered *A. seminudus* a synonym of *C. sistrata* (Fr.) Singer ex Bon & Bellù [10,15,30,31,32,33], while other authors regarded them as different species [8,20,34,35]. Fayod even erected the genus *Fusispora* Fayod to accommodate *L. sistrata* and considered *A. seminudus* as a member of another new genus, *Cystoderma* [25]. In contrast, Vellinga reduced *L. sororia* Huijsman and *L. rufipes* Morgan to synonymy with *C. seminuda* [21,36].

The current application of the name *C. seminuda* is unstable. Preliminary phylogenies with ITS sequence data retrieved from GenBank placed specimens identified in previous studies as *C. seminuda* [18,37,38] in different clades within the genus *Cystolepiota*. During our investigations of the Agaricaceae in recent years, we compiled several collections with fruit bodies closely resembling *C. seminuda* in overall morphology but exhibiting a considerable degree of variation. Observations under scanning electron microscope (SEM) revealed distinct differences in spore ornamentations, while further analyses of DNA sequences also exhibited significant molecular differences, indicating the existence of more species in addition to *C. seminuda*.

The aim of this study was therefore to: (1) re-delimitate *Cystolepiota* and discuss its relationships with related genera; (2) typify and circumscribe *C. seminuda* based on recent specimens collected close to its type locality and clarify its systematic position; and (3) explore the species diversity of *Cystolepiota* and describe novel species based on morphological characteristics and DNA sequence data.

## 2. Materials and Methods

### 2.1. Morphological Studies

Macroscopic descriptions are based on field notes and digital images of fresh basidiomata. The color code of Kornerup and Wanscher was used [39]. Measurements and descriptions of microscopic characteristics were performed on dried materials. Sections of specimens were rehydrated in 5% aqueous (*w*/*v*) potassium hydroxide (KOH) and stained with 1% Congo red reagent, if necessary. Microscopic features were observed and measured under a Leica DM2500 light microscope (LM). Basidia, basidiospores, cystidia, and pileal structures were measured and depicted using brightfield or phase contrast optics. Chemical reactions of basidiospores were tested in Melzer’s reagent and cresyl blue. The lamella fragments of dried specimens were fixed on an aluminum pile with double-sided adhesive tape, then coated with gold palladium and observed under a ZEISS Sigma 300 scanning electron microscope (SEM).

The notations [n/m/p] indicate that the measurements were taken on n basidiospores, from m basidiomata and p collections. The dimensions of basidiospores were provided with the notation (a–)b–c(–d). The range b–c indicates the minimum and maximum values of 90% of the measured structures. The a and d in parentheses indicate the absolute extreme values. Q is used to indicate the length/width ratio of basidiospores in lateral view; Q_m_ means average Q of all basidiospores ± sample standard deviation.

The spore size data from the type material of *Lepiota sororia* was obtained from Figure 1 in Vellinga (1987) using WebPlotDigitizer (https://automeris.io/WebPlotDigitizer/ (accessed on 5 September 2022)) [21]. The scatter diagrams were plotted in R v4.2.1 using the ggplot2 package [40,41].

Specimens are deposited in the Herbarium of Cryptogams, Kunming Institute of Botany, Chinese Academy of Sciences (KUN) and in the fungarium of the Senckenberg Museum of Natural History, Görlitz (GLM).

### 2.2. DNA Extraction, PCR Amplifications and Sequencing

Genomic DNA was extracted from dry basidiomata using a fungal genomic DNA extraction kit (BioTeke Corporation, Beijing, China) following the manufacturer’s instructions. Four DNA fragments were amplified and sequenced: ITS, LSU, the most variable region of the second-largest subunit of RNA polymerase II (*rpb2*), and a portion of the translation–elongation factor 1-α (*tef1*). The primer pairs ITS1F/ITS4 [42,43], LR0R/LR5 [44,45], bRPB2-6F/bRPB2-7.1R [46], and EF1-983F/EF1-1567R [47] were used for the amplification of the respective DNA fragments. PCR amplification and sequencing protocols followed those of Hou and Ge (2020) [18]. The 50 newly-generated sequences were deposited in GenBank (Appendix A).

### 2.3. Phylogenetic Analyses

The phylogenetic analyses included 120 ITS, 57 LSU, 44 *rpb2*, and 42 *tef1* sequences. Based on recent phylogenetic studies [18,19], sequences of three *Echinoderma* species were chosen as the outgroup of the ITS analysis and sequences of two *Coprinus* species for the combined analysis. Multiple sequences alignments were performed with MAFFT v7.4.95 [48] and ambiguously aligned regions of each sequence were detected and excluded using trimAl v1.4 [49]. The resulting alignments were examined and optimized manually in AliView 1.27 [50].

Firstly, maximum likelihood (ML) phylogenetic trees were reconstructed based on matrices of each respective gene fragment, viz. the ITS matrix (120 sequences), the subsampled ITS matrix with selected sequences, the LSU matrix, the *rpb2* matrix, and the *tef1* matrix. As no conflicts with high support (>50%) were observed among the ML trees of the five data sets (Figure 1 and Appendix A), three multi-locus concatenated matrices (ITS-LSU-*rpb2-tef1*, LSU-*rpb2-tef1,* and *rpb2-tef1* matrix) were constructed using PhyloSuite v1.2.2 [51] and analyzed by the ML method and bayesian inference (BI) method.

The best-fit substitution model for each single alignment was estimated by PartitionFinder 2 [52]. The BI analyses were performed with MrBayes v. 3.2.6 [53] by applying the best-fit model for each partition (2 parallel runs, 10,000,000 generations), in which the initial 25% of sampled data was discarded as burn-in. The Bayesian posterior probabilities (BPPs) were calculated from the posterior distribution of the remaining phylogenetic trees. ML estimation was conducted with RAxML 8.2.12 [54] using GTR+I+GAMMA as the best-fit likelihood model. Statistical supports were calculated with 1000 bootstrap replicates. Resulting phylograms were displayed in FigTree 1.4.4 (http://tree.bio.ed.ac.uk/software/figtree/ (accessed on 2 November 2021)).

## 3. Results

### 3.1. Phylogenetic Results

The ITS data set consisted of 120 sequences and 745 sites, including 356 parsimony informative sites. The final matrix of the combined data set included 89 ITS, 57 LSU, 44 *rpb2*, and 42 *tef1* sequences, in which the aligned lengths of the four gene fragments were 651, 862, 705, and 563 bp, respectively, that contained 963 parsimony informative sites. As the ITS phylogenetic tree included the largest number of samples and the ITS-LSU-*rpb2-tef1* phylogenetic tree was the second-most supported phylogenetic tree with relatively more species than other phylogenetic trees inferred from the combined data set (Figure 1, Figure 2, and Appendix A), the phylogenies based on the ITS data set and the combined ITS-LSU-*rpb2-tef1* data set were displayed and used for subsequent discussions (Figure 1 and Figure 2).

In the ITS phylogeny (Figure 1), most of the specimens previously identified as *C. seminuda* and either sequenced in this study or from previous studies belong to Clade I (ML bootstrap [MLB] = 66%, Bayesian posterior probability [BPP] = 0.94) or Clade II (MLB = 98%, BPP = 1.00); both together represent the *C. seminuda* complex. Clade I consists of a clade formed by several subclades, including *C. pseudoseminuda* (MLB = 100%, BPP = 1.00) and *C.* aff. *pseudoseminuda* 1 (MLB = 100%, BPP = 1.00) and 2 (MLB = 100%, BPP = 1.00), two specimens previously identified as *Melanophyllum eyrei* (Massee) Singer that grouped with low support (MLB = 52%, BPP < 0.90) and several further unnamed subclades. Clade II includes *C. seminuda* (s. str.) and several further unnamed subclades as well. The *C. seminuda* (s. str.) subclade is the clade with the largest number of specimens previously identified as this species and is well supported (MLB = 97%, BPP = 0.99). Two specimens from Germany (GLM-F107803 and GLM-F107804) previously identified as *C. seminuda* form a distinct separate clade (MLB = 100%, BPP = 1.00). One specimen from China (KUN-HKAS 53985) forms another distinct clade (MLB = 100%, BPP = 1.00) with a specimen from Italy previously identified as *C. seminuda* and an uncultured *Cystolepiota* clone from India. A further specimen from China (KUN-HKAS 124759) is grouped with the holotype of *C. oliveirae* P. Roux, M. Paraíso, J.-P. Maurice, A.-C. Normand and F. Fouchier (MLB = 88%, BPP = 0.98) and forms a sister clade to two specimens of an unnamed species from Hawaii. Both are grouped with *Cystolepiota petasiformis*, representing the well-supported *Pulverolepiota* clade (MLB = 100%, BPP = 1.00). Specimens labeled *C. bucknallii*, *C. icterina*, and *C. rhodella* Sysouph. and Thongkl in GenBank, which represent three species with dextrinoid basidiospores, does not form a monophyletic group.

In the phylogeny inferred from ITS-LSU-*rpb2*-*tef1* (Figure 2), five main clades can be recognized: the clade including *Cystolepiota* and *Melanophyllum* (MLB = 77%, BPP = 0.91), the *Echinoderma* clade (MLB = 100%, BPP = 1.00), the *Smithiomyces* clade (MLB = 100%, BPP = 1.00), the *Lepiota* clade (MLB = 63%, BPP = 1.00), and the *Pulverolepiota* clade (MLB = 100%, BPP = 1.00). Most of the specimens previously identified as *C. seminuda* and either sequenced in this study or from previous studies belong to Clade I (MLB = 75%, BPP = 0.98) or Clade II (MLB = 100%, BPP = 1.00) in the first main clade (*Cystolepiota*/*Melanophyllum* clade). The two clades of the *C. seminuda* complex are sister to each other, forming a clade with low support (MLB = 59%, BPP < 0.90). Clade I consists of *C. pseudoseminuda*, *C.* aff. *pseudoseminuda* 1 and 2, *M. eyrei*, and several further unnamed subclades. Clade II includes *C. seminuda* (s. str.) and several further unnamed subclades as well. One specimen from China (KUN-HKAS 53985) forms a distinct clade (MLB = 100%, BPP = 1.00) with a specimen from Italy previously identified as *C. seminuda* and an uncultured *Cystolepiota* clone from India within the *Cystolepiota*/*Melanophyllum* main clade. Two specimens (GLM-F107803 and GLM-F107804) from Germany also form a distinct clade (MLB = 100%, BPP = 1.00) within the *Cystolepiota*/*Melanophyllum* clade. A further specimen from China (KUN-HKAS 124759) is grouped with the holotype of *C. oliveirae* within the *Pulverolepiota* clade. 

The *C. seminuda* complex formed a well-supported clade in all the resulting phylogenetic trees based on *rpb2*, *tef1*, as well as the combined *rpb2-tef1* data set (Appendix A). However, the ITS-LSU-*rpb2-tef1* combined data set, and LSU-*tef1-rpb2* combined data set support the monophyly of the *C. seminuda* complex with low supports, while the LSU data set showed the lowest signals for the phylogeny of *Cystolepiota*. The presence of ITS and/or LSU in the combined data sets seem to decrease the statistical support of certain nodes of the phylogenetic tree inferred from the combined data set (Figure 1, Figure 2 and Appendix A).

### 3.2. Taxonomy

***Cystolepiota pseudoseminuda*** Y.J. Hou, H. Qu & Z.W. Ge, **sp. nov.** Figure 3A–C, Figure 4C,D and Figure 5.

**MycoBank**: MB845494.

**Etymology**: The name refers to the resemblance of the new species to *Cystolepiota seminuda* [Latin pseudo = false].

**Diagnosis**: *Cystolepiota pseudoseminuda* is distinguished from other *Cystolepiota* species by its slender basidiomata, pulverulent or granulose squamules with a pink to orange tinge that are composed of loosely-arranged inflated cells connected to filamentous hyphae, non-dextrinoid basidiospores with distinct warts on the surface visible under LM and SEM, the absence of cystidia, and the presence of abundant clamp connections. Its ITS, LSU, *rpb2*, and *tef1* sequences differentiate this species from other species.

**Type:** China, Yunnan Province, Kunming City, Kunming Botanical Garden, on soil under Cupressaceae, alt. 1980 m, 21 October 2015, Z.W. Ge 3795 (**Holotype**: KUN-HKAS 92275). GenBank: ITS = MN810149, LSU = MN810101, *rpb2* = MN820980, *tef1* = MN820926.

**Description:** Pileus 5–18 mm diam, hemispherical to obtusely conical when young, expanding to plano-convex or applanate with a slightly umbonate center with age; surface dry, white to cream, with pulverulent to granulose squamules, fugacious, white, cream, pale orange (5A3), pink (10A3), pinkish-orange to light pink (6A2-5) at the center, turning white to cream towards the margin; margin of the pileus with easily detachable veil remnants. Lamellae up to 2(3) mm broad, free, moderately crowded, unequal, white to light cream, with 1–3 tiers of small lamellulae. Stipe 20–50 × 0.5–2 mm, central, subcylindrical to cylindrical, surface white to cream on the upper portion, with age gradually turning to pale orange (5A3-4), grayish orange (6B5-7), brownish (5D6) to purplish brown (7F7) towards the middle and lower portion with age, with pulverulent to granulose squamules, which are concolorous with those of the pileus; with a fragile and fugacious membranous veil between pileus margin and stipe initially; basal mycelium white. Context thin, whitish. Odorless; taste not recorded.

Basidiospores [100/5/3] (3–)3.5–4.5(–5) × 2–3 (–3.5) μm, Q = (1.21–)1.24–1.85(–2.20), Q_m_ = 1.55 ± 0.19, ovoid to ellipsoid, colorless, thin-walled, inamyloid, non-dextrinoid, metachromatic in cresyl blue; surface slightly punctate-rough under the LM, distinct warts (up to 0.15 µm high) visible under SEM (Figure 4C,D); apiculus small. Basidia 13–23 × 3.5–8.5 μm, clavate, hyaline, 4-spored, rarely 2-spored; sterigmata 1–3 μm long. Lamellar trama regular, made up of cylindrical colorless hyphae, 3–15 μm in diam. Cheilocystidia and pleurocystidia absent. Squamules composed of loosely-arranged globose, subglobose, or ellipsoid, rarely sphaeropedunculate cells, 17–28 × 15–24 µm, smooth-walled, slightly thick-walled, with colorless or sometimes yellowish intracellular pigments; the abovementioned cells are usually attached to hyaline hyphae, 1–6 µm in diam, colorless. Clamp connections present in all tissues (Figure 5).

**Habitat and distribution:** solitary or scattered on nutrient-rich soil or rotten leaves, distributed in temperate and subtropical zones of southwestern and central China.

**Additional specimens examined:** China, Gansu Province, Longnan City, Wen County, Chengguan Town, hill near Jiachang Village, on rotting leaves under bushes (Fagaceae, Rosaceae and *Coriaria nepalensis*), alt. 1700 m, 26 August 2011, X.T. Zhu 574 (KUN-HKAS 73969); Yunnan Province, Kunming City, Kunming Botanical Garden, alt. 1915 m, 9 October 2005, Z.W. Ge 923 (KUN-HKAS 49482).

**Notes:** *Cystolepiota pseudoseminuda* is morphologically similar to *C. seminuda* regarding color and size of the basidiomata. Both species form pulverulent to granulose squamules, non-dextrinoid basidiospores, and clamp connections, while cystidia are lacking. However, the surface of basidiospores of *C. seminuda* is smooth under LM and SEM, which is in contrast to the warty basidiospores of *C. pseudoseminuda*. The basidiospores of *C. pseudoseminuda* are shorter and wider at Q = (1.21–)1.24–1.85(–2.20) and Q_m_ = 1.55 ± 0.19, compared to those of *C. seminuda* at Q = (1.41–)1.46–2.15(–2.45) and Q_m_ = 1.78 ± 0.22. At least according to the specimens examined in this study, although the characteristics of basidiospores vary strongly between specimens of *C. pseudoseminuda* and *C. seminuda* and there is an overlap between the two species in basidiospore size and shape, individuals with high Q_m_ value always belong to *C. seminuda* and those with low Q_m_ value always belong to *C. pseudoseminuda* (Figure 6). In addition, there are 92 (out of 722) nucleotide differences between the ITS sequences of the holotype of *C. pseudoseminuda* and that of the neotype of *C. seminuda.*

Specimens of *C. pseudoseminuda* form a robust clade with *C.* aff. *pseudoseminuda* 1 and *C.* aff. *pseudoseminuda* 2 in in both phylogenies (Figure 1 and Figure 2). The ITS sequences of the three groups are 96% to 97% identical. Their geographical distribution patterns are different: *C. pseudoseminuda* has so far been collected only in southwestern China and *C.* aff. *pseudoseminuda* 1 in Europe and northern China, while all the sequences of *C.* aff. *pseudoseminuda* 2 are derived from specimens collected in the Unites States of America (USA). Based on the observation of specimens in this study, no distinct morphological differences were found to distinguish *C. pseudoseminuda* and *C. pseudoseminuda* 1 (Figure 4A–D) and no specimens of *C. pseudoseminuda* 2 were studied morphologically. Further morphological observations need to be conducted on *C.* aff. *pseudoseminuda* 1 and *C.* aff. *pseudoseminuda* 2.

***Cystolepiota pyramidosquamulosa*** H. Qu & Z. W. Ge **sp. nov.**
Figure 3I, Figure 4L and Figure 7.

**MycoBank**: MB845060.

**Etymology**: The epithet “pyramidosquamulosa” refers to the pyramidal squamules on the pileus.

**Diagnosis**: *Cystolepiota pyramidosquamulosa* is distinguished from all other *Cystolepiota* species by its brownish, relatively large, irregular pyramidal squamules on the pileus, an annulus-like zone on the upper portion of the stipe, the smooth-walled (under LM and SEM), inamyloid, and non-dextrinoid basidiospores, the absence of cystidia, and the presence of clamp connections. Its ITS, LSU, *rpb2*, and *tef1* sequences are distinct from other species.

**Type**: China, Sichuan Province, Aba Tibetan and Qiang Autonomous Prefectures, Maerkang (Barkam) City, Dangba Township, Yinlang Village, on soil, 17 August 2007, Z. W. Ge 1900 (**Holotype**, KUN-HKAS 53985). GenBank: ITS = OP059088, LSU = OP059068, *rpb2* = OP104335, *tef1* = OP141792.

**Description**: Pileus 24–28 mm in diam, convex to plano-convex with an indistinct broad umbo; surface white, covered with concolorous flocculose squamules and brownish (5B4-5) irregular pyramidal squamules (up to 5 mm in height) that spread out towards the margin and can easily be wiped off; margin with overhanging squamules similar to those on the pileus. Lamellae free, close, up to 2.5 mm wide, yellowish-white (2A3), with 1–3 tiers of lamellulae; lamellae margin smooth. Stipe 30–45 × 15–25 mm, central, cylindrical, with a slightly expanded base, hollow, stuffed with white fibers, brownish (5B4-5); glabrous or slightly pruinose at the apex, with an annulus-like zone on the upper part covered with similar squamules as the pileus; squamules on lower portion cream to yellowish white (2A2-3), flocculose or pulverulent. Context of stipe and pileus cream to brownish (5B4). Smell and taste were not recorded.

Basidiospores [100/5/1] (3.5–)4–5 (–5.5) × (1.5–)2–3 μm, Q = (1.44–)1.60–2.28(–2.47), Q_m_ = 1.95 ± 0.20, ellipsoid to elongate, sometimes cylindrical, colorless, thin-walled, smooth-walled under the LM and SEM (Figure 4L); inamyloid, non-dextrinoid, metachromatic in cresyl blue. Basidia (13.5–)15–18.5 × 5–7 μm, clavate, hyaline, 4-spored; sterigmata 2.5–3.5 μm long. Cheilocystidia and pleurocystidia not observed. Lamellar trama regular, colorless, composed of cylindrical hyphae, 5–7.5 μm in diam. The squamules on pileus and stipe consist of inflated cells and filamentous hyphae; inflated cells abundant, globose to subglobose, rarely fusiform or sphaero-pedunculate, 21–33 × 20–29 µm, smooth-walled, with a slightly thick, chartreuse-yellow wall, usually 2–5 cells forming loosely arranged chains; hyphae rare, 1–3 µm in diam, chartreuse-yellow. Clamp connections present in all tissues (Figure 7).

**Habitat and distribution:** Presumably saprotrophic. Solitary or scattered on soil. Found in Asia (China, India) and Europe (Italy).

**Notes:** Pyramidal squamules are the most distinctive macroscopic characteristic of *Cystolepiota pyramidosquamulosa*. The pilei of *P. oliveirae* and *P. petasiformis* (=*P. pulverulenta*) are also covered with obvious squamules and also lack cystidia. However, the two species can be distinguished from *C. pyramidosquamulosa* by the rough basidiospores that slowly become red-brown in Melzer’s reagent and the squamules composed of irregular, branched, inflated, and oblong elements. In addition, the basidiospores of *C. pyramidosquamulosa* are narrower than those of the latter two species [5,55,56]. *Cystolepiota pyramidalis* Sysouph. and Thongkl. forms squamules similar to *C. pyramidosquamulosa.* However, *C. pyramidalis* forms cheilocystidia and ellipsoid–ovoid spores and has so far only been collected in Laos and Thailand [57]. In contrast to *C. pyramidosquamulosa*, *C. fumosifolia* (Murrill) Vellinga and *C. pseudofumosifolia* M.L. Xu & R.L. Zhao form warty to granulose squamules and have cystidia [5,58]. *Cystolepiota adulterina* F.H. Møller ex Bon has warty squamules and somewhat resembles *C. pyramidosquamulosa* macroscopically. However, it differs in the presence of abundant cheilocystidia. In addition, *C. pyramidosquamulosa* can be distinguished from other species by its distinct ITS, LSU, *rpb2*, and *tef1* sequences.

Two ITS sequences from GenBank, JF907983 (from Italy) and KU847887 (from India), are similar to the sequence derived from the type specimen of *C. pyramidosquamulosa*; both differ by seven nucleotides from *C. pyramidosquamulosa*, which is considered within the variability of *C. pyramidosquamulosa*.

***Cystolepiota seminuda*** (Lasch) Bon, Documents Mycologiques 6(24): 43, 1976. Figure 3D–F, Figure 4E–G and Figure 8.

≡ *Agaricus seminudus* Lasch, Linnaea 3: 157, 1828 (Basionym).

≡ *Lepiota seminuda* (Lasch) P. Kumm., Führ. Pilzk. (Zerbst): 136, 1871.

≡ *Lepiota sistrata* var. *seminuda* (Lasch) Quél., Mém. Soc. Émul. Montbéliard, Sér. 2 5: 231, 1872.

≡ *Cystoderma seminudum* (Lasch) Fayod, Annls Sci. Nat., Bot., sér. 7 9: 351, 1889.

≡ *Mastocephalus seminudus* (Lasch) Kuntze, Revis. gen. pl. (Leipzig) 2: 860, 1891.

= *Lepiota seminuda* f. *minima* J.E. Lange, Fl. Agaric. Danic. 1: 36, 1935 (Nom. inval.).

= *Lepiota sistrata* f. *minima* J.E. Lange ex Babos, Annls hist.–nat. Mus. natn. hung. 50: 91, 1958 (Nom. inval.).

**Neotypification:** Germany, Saxony, Zwickau, Helmsdorf-Nord, edge of the forest, 10 October 2007, H. Jurkschat (GLM-F125824, **neotype** designated here), MycoBank: MBT10008384, GenBank: ITS = OL898735.

**Description**: Pileus 2.5–12 mm diam, first conical, paraboloid to hemispherical, then expanding to plano-convex or convex, often with a broad umbo at the center; surface white, when young, changing to light cream with age, covered with densely powdery to granulose squamules, which become sparse or even disappear with age; light yellowish-brown (6C3-5), pinkish-orange to light pink (6A2-5) at the center of the pileus; white to cream towards the margin, fragile and easily breaking off; pileus margins often appendiculate with veil remnants. Lamellae up to 2(–3) mm wide, free, crowded, white to light cream, with 3–5 tiers of lamellulae. Stipe 10–45 mm long, 0.5–1.5 mm diam, central, subcylindrical to cylindrical; white to cream; glabrous in the upper part, gradually darkening towards the base, central and lower part grayish orange (6B5-7), brownish (5D6), reddish brown (7C6-7) to purplish brown (7F7), with white, light cream to light brownish powdery to granulose squamules, easily breaking off. Context thin (<1 mm), whitish. Smell not distinct; taste not recorded.

Basidiospores [160/8/7] (3–)3.5–4.5(–5) × (1.5–)2–2.5(–3) μm, Q = (1.41–)1.46–2.15(–2.45), Q_m_ = 1.78 ± 0.22, ellipsoid to elongate; inamyloid, non-dextrinoid, metachromatic in cresyl blue; colorless, surface smooth under LM and SEM (Figure 4E–G); apiculus small. Basidia (11–)13–18.5(–20) × (4–)5–6(–7) μm, clavate, 4-spored; sterigmata 1.5–3 μm long. Lamellar trama regular, composed of hyphae, 3–9 μm diam. Pleurocystidia and cheilocystidia absent. Squamules composed of loosely arranged globose, subglobose, ellipsoid, or sometimes sphaeropedunculate cells, 16–41 × 14–35 µm and slender hyphae, 1–4 µm in diam, smooth-walled, thick-walled, colorless. Clamp connections present in all tissues (Figure 8).

**Habitat and distribution:** solitary or scattered on soil, rotten leaves, or between moss, distributed in temperate and subtropical zones of the Northern Hemisphere (China, Estonia, Germany, USA).

**Additional specimens examined:** China, Yunnan Province, Kunming City, Golden Temple, 14 June 2008, Z.W. Ge 2014 (KUN-HKAS 54210); ibid., Z.W. Ge 2015 (KUN-HKAS 54211); Kunming Botanical Garden, 28 October 2017, Z.W. Ge 4102 (KUN-HKAS 106016); Xiaoshao, 25 October 2017, Z.W. Ge 4094 (KUN-HKAS 106008); Xizang (Tibet) Autonomous Region, Qamdo Prefecture, Mangkang (Markam) County, alt. 3302 m, 4 August 2013, Z.W. Ge 3437 (KUN-HKAS 84275). Estonia, Viljandi S, Loodi, 17 August 1989, G. Zschieschang (GLM-F040567); Heimtali, 17 August 1989, G. Zschieschang (GLM-F040540). Germany, North Rhine-Westphalia, Lohmar, Wahner Heide, 1 November 2016, K. Wehr (GLM-F109912); Saxony, Görlitz-Biesnitz, Landeskrone, under *Fagus*, 15 September 1996, G. Zschieschang (GLM-F036422); Saxony Herrnhut, Hutberg, under *Tilia* and *Quercus*, 27 September 2001, G. Zschieschang (GLM-F042189).

**Notes:** *Agaricus seminudus*, the basionym of *Cystolepiota seminuda*, was described in the Margraviate of Brandenburg (not identical with the contemporary state of Brandenburg), Germany [23]. As no holotype specimen of *A. seminudus* was designated, specimens previously collected in Germany that had been identified as *C. seminuda* based on morphology were studied. The ITS sequences of these specimens confirmed the placement of this species in the genus *Cystolepiota*. However, these specimens belonged to four clades, suggesting the existence of four species. Two of them clustered with sequences from GenBank that had been generated from specimens identified as *C. seminuda* as well and represent two different species. The basidiospores of the specimens of one species were observed to be ornamented under the SEM, while those of another species were smooth-walled. We regard the latter species as *C. seminuda* because it includes specimens that had been collected closest to the type location and most of the German specimens belonged to it. Moreover, these specimens were presumably growing on soil, two specimens from Estonia were even collected between moss described as the habitat of *A. seminudus* in the protologue of the species. As the holotype was neither located in ZE Botanischer Garten und Botanisches Museum, Freie Universität Berlin (B), where the type specimens of G. W. Lasch had been deposited, nor in other collections holding specimens collected by Lasch (https://www.sil.si.edu/DigitalCollections/tl-2/history.cfm (accessed on 1 December 2022)), one of the specimens closest to the type location (GLM-F125824) is designated here as a neotype of *A. seminudus*. In contrast, the two German specimens from the clade with ornamented basidiospores were apparently growing on wood and related to *C. pseudoseminuda,* which is a species newly described in this study.

Compared with *C. seminuda*, *L. sororia* and *L. rufipes*, which have been reduced to synonyms of *C. seminuda* by Vellinga in 1987 and 2010 respectively, have larger basidiomata and longer and narrower basidiospores [21,33,36,59]. Furthermore, *L. sororia* forms larger basidiomata [33] and both the pileus and stipe of *L. rufipes* were described as smooth and glabrous [59], although Vellinga (2010) inferred that it may be the result of rain washing [36]. Accounting for the difference described above, we temporarily remove *L. sororia* and *L. rufipes* from synonyms of *C. seminuda* until specimens exactly matching the description of the protologue of the two species are found and sequenced.

***Pulverolepiota*** Bon, Docums Mycol. 22(no. 88): 30, 1993.

≡ *Leucoagaricus* sect. *Pulverulenti* Bon, Docums Mycol. 8(nos 30-31): 71, 1978.

≡ *Cystolepiota* sect. *Pulverolepiota* (Bon) Vellinga, Persoonia 16: 525, 1998.

**Type**: *Lepiota petasiformis* Murrill, Mycologia 4(5): 232, 1912.

**Notes:** Bon (1978) transferred *Lepiota pulverulenta* to *Leucoagaricus* Locq. ex Singer and erected *Leucoagaricus* sect. *Pulverulenti* Bon typified by *Leucoagaricus pulverulentus* (Huijsman) Bon [60]. In 1993, he erected a new genus, *Pulverolepiota*, to accommodate this species [4]. Vellinga and Huijser (1998) subsequently reduced *Pulverolepiota* to a section of *Cystolepiota*, *C.* sect. *Pulverolepiota* [6]. However, specimens previously identified as *C. petasiformes*, *C. oliveirae*, and related species representing *Cystolepiota* sect. *Pulverolepiota* form a distinct clade in both phylogenies (Figure 1 and Figure 2). The respective clade in Figure 2 is one of the main clades that represent genera and did not group with the *Cystolepiota*/*Melanophyllum* clade, but is sister to the clades formed by *Cystolepiota*, *Melanophyllum*, *Echinoderma*, *Smithiomyces*, and *Lepiota*., supporting *Pulverolepiota* as being a distinct genus rather than a section of *Cystolepiota*. Therefore, the genus *Pulverolepiota* Bon (1993) is resurrected here.

***Pulverolepiota oliveirae*** (P. Roux, M. Paraíso, J. P. Maurice, A. C. Normand & F. Fouchier) H. Qu, Damm & Z. W. Ge, **comb. nov.** Figure 3G,H, Figure 4J,K and Figure 9.

≡ *Cystolepiota oliveirae* P. Roux, M. Paraíso, J. P. Maurice, A. C. Normand & F. Fouchier, Mycologia Montenegrina 19: 22 2017 (Basionym).

**MycoBank**: MB847958

**Description**: Pileus 14 mm in diam, about 18 mm high, plano-conical, with a distinct umbo, surface whitish to cream, covered with thick flocculose squamules, erect at center, brownish orange (6C8) at the center, fading to yellowish or pale orange (5A2-4) or cream toward the margin; margin appendiculate, easily detachable. Lamellae free, wavy, brownish (5B2), distant, up to 2 mm broad, with 2–3 tiers of lamellulae. Stipe 30 × 1.5 mm, light orange (5A5) at the upper part, reddish-orange (7A6-8) at the base, central, hollow, subcylindrical, with a slightly bulbous base; surface smooth, covered with white, flocculose, easily-detachable squamules on the middle part. Context white in stipe and pileus, thin. Odorless; taste not recorded.

Basidiospores [30/1/1] (4–)4.5–5.5(–6) × (2.5–)3–3.5 μm, Q = (1.43–)1.46–1.86(–2.03), Q_m_ = 1.66 ± 0.14, ellipsoid, colorless, thin-walled, nonamyloid, slowly staining to reddish-brown in Melzer’s reagent (after 24 h); metachromatic in cresyl blue; surface distinctly punctate-rough under LM; isolated irregular warts (up to 0.3 µm in height) visible under SEM (Figure 4J,K); apiculus small. Basidia 14.5–18.5 × 5–7 μm, clavate, hyaline, 4-spored; sterigmata 1.5–3 μm long. Lamellar trama regular, made up of colorless cylindrical hyphae, 3–10 μm in diam. Cheilocystidia and pleurocystidia not observed. Squamules mainly composed of loosely arranged, irregular, branched, cylindrical, oblong cells with a brownish, slightly thick wall; rare tightly-arranged globose to subglobose cells present in the midst of loosely arranged cells, 36–55 × 40–70 μm, colorless, covered with thick gelatinous substances. Clamp connections absent (Figure 9).

**Specimen examined:** China, Yunnan Province, Xishuangbanna Dai Autonomous Prefecture, Jinghong City, Xishuangbanna Primitive Forest Park, on the roadside, 100.8837777° N, 22.0348152° E, alt. 718 m, 5 July 2021, H. Qu 375 (KUN-HKAS 124759).

**Habitat and distribution:** Solitary or scattered on soil or on a trunk of *Dicksonia antarctica* in Europe (Portugal), Asia (tropical region of southwestern China) and Australia.

**Notes:***Pulverolepiota oliveirae* was described as *C. oliveirae* from Portugal [55], characterized by a whitish to cream pileus with erect flocculose squamules, a brown stipe with white flocculose squamules, a lack of cystidia, and rough and slow-staining dextrinoid spores. Our collection represents a new report for China; it matched the protologue of *P. oliveirae* regarding macro- and micro-characteristics, except for the pileus of the specimen from China that has a distinct umbo. The basidiospores of the specimen from China [(4–)4.5–5.5(–6) × (2.5–)3–3.5 µm] are slightly smaller than those in the protologue of *C. oliveirae* (5.0–6.4 × 2.9–4.0 µm) [55]. The morphological differences could be explained by population and measurement differences. Furthermore, the ITS sequence is 99.74% identical with the sequence generated from the type specimen (KY472789).

The two closely related species, *P. oliveirae* and *P. petasiformis* (=*C. pulverulenta*) can easily be confused. However, the squamules of *C. oliveirae* present tightly-arranged globose to subglobose cells among the loosely-arranged cells. Those tightly arranged cells have not been observed in *C. petasiformis* [5,56]. Paraíso et al. (2016) suggested the size of the basidiospores as a characteristic to distinguish the two species [55]. However, the spores of the collection from China are not significantly larger than those of *C. petasiformis*. Several *Cystolepiota* species, including *C. fumosifolia*, *C. pseudofumosifolia,* and *C. pyramidalis*, that are not closely related with *P. oliveirae* are also similar in the presence of distinct squamules. However, they differ in forming cheilocystidia, inamyloid and non-dextrinoid basidiospores and squamules that are composed of loosely-arranged globose to subglobose cells, while irregular, branched, cylindrical to oblong cells are lacking. *Cystolepiota pyramidosquamulosa*, a species newly described in this paper, also forms distinct squamules. It can be distinguished from *P. oliveirae* by having neither amyloid nor dextrinoid basidiospores and forming loosely-arranged globose to subglobose cells in the squamules.

***Pulverolepiota petasiformis*** (Murrill) H. Qu, Damm & Z. W. Ge, **comb. nov.**

≡ *Lepiota petasiformis* Murrill, Mycologia 4(5): 232, 1912 (Basionym).

≡ *Cystolepiota petasiformis* (Murrill) Singer, Agaric. mod. Tax., Edn 2 (Weinheim): 472, 1975. Nom. inval., Art. 35.2 (Melbourne).

≡ *Cystolepiota petasiformis* (Murrill) Vellinga, Mycotaxon 98: 228, 2006.

= *Lepiota pulverulenta* Huijsman, Persoonia 1(3): 328, 1960.

≡ *Leucoagaricus pulverulentus* (Huijsman) Bon, Docums Mycol. 8(nos 30-31): 70, 1978.

≡ *Pulverolepiota pulverulenta* (Huijsman) Bon, Docums Mycol. 22(no. 88): 30, 1993.

≡ *Cystolepiota pulverulenta* (Huijsman) Vellinga, Persoonia 14(4): 407, 1992.

≡ *Leucoagaricus pulverulentus* var. *subroseus* Bon, Docums Mycol. 8(nos 30-31): 71, 1978.

≡ *Pulverolepiota pulverulenta* var. *subrosea* (Bon) Bon, Docums Mycol. 22(no. 88): 30, 1993.

= *Leucoagaricus pulverulentus* f. *minimus* Bon, Migl. & Brunori, Docums Mycol. 19(no. 75): 54, 1989.

= *Pulverolepiota pulverulenta* f. *minima* (Bon, Migl. & Brunori) Bon, Docums Mycol. 22(no. 88): 30, 1993.

= *Cystolepiota pulverulenta* f. *minima* (Bon, Migl. & Brunori) La Chiusa, Riv. Micol. 41(2): 152, 1998.

**MycoBank**: MB847957

**Notes**: *Pulverolepiota petasiformis* was originally described by Murrill (1912) from Washington State, USA as *Lepiota petasiformis* [61]. Vellinga (2006) transferred it to *Cystolepiota* after examining its type specimen. Because of the similar morphology, for example, *L. petasiformis* also lacks clamp connections, she treated the later described *L. pulverulenta*, a European species and the type species of *Pulverolepiota*, as a synonym of *C. petasiformis*, while morphological differences exist as well [5]. Moreover, in molecular analyses of a previous study, sequences from Europe and North America formed a well-supported monophyletic clade (Figure 1, Figure 2, Figure 3 in Vellinga 2003) [17]. As the clade including *C. petasiformis* represents a generic level clade named *Pulverolepiota* (Figure 2), a new combination *Pulverolepiota petasiformis* is proposed here.

A total of five sequences of *P. petasiformis* were included in the ITS phylogeny (Figure 1), including three sequences of collections originally identified as *C. pulverulenta* by Vellinga [62] and two sequences from GenBank labeled as *Cystolepiota* sp. The five sequences formed a well-supported clade (MLB = 94%, BPP = 1.00) with a subclade (MLB = 97%, BPP = 1.00) formed by two European sequences. As can be seen in Figure 1, there are large differences between the European and the North American collections, indicating them to be distinct populations that may result from geographical isolation and may represent *P. pulverulenta* and *P. petasiformis* respectively. However, there is also a large variability within the specimens from North America. Further collections and studies are needed to evaluate, if *P. petasiformis* is one variable species or a species complex.

## 4. Discussion

### 4.1. Circumscription and Diversity of Cystolepiota and Pulverolepiota

According to the phylogenetic results of this study (Figure 1 and Figure 2), the resurrected genus *Pulverolepiota* includes *P. petasiformis*, *P. oliveirae* and an unnamed species from Hawaii. The main characteristics of *Pulverolepiota* are: rough basidiospores that are slowly staining reddish-brown in Melzer’s reagent, the absence of clamp connections and cystidia, and pileus-covering elements that are composed of loosely-arranged elongate, branched, irregular, sometimes subglobose or ellipsoid cells. *Lepiota roseolanata* [≡*Leucoagaricus roseolanatus* (Huijsman) Bon], a species described by Bon (1962) based on a specimen from Limburg, Netherlands, matches the above characteristics and may be a member of *Pulverolepiota*. Vellinga noted that *Cystolepiota pseudogranulosa* (Berk. & Broome) Pegler may belong to *Pulverolepiota*, while further studies are needed as *C. pseudogranulosa* forms strongly dextrinoid basidiospores and clamp connections [6,9].

With the exclusion of *C.* sect. *Pulverolepiota*, two sections remain in *Cystolepiota*: *C.* sect. *Cystolepiota* that include species with non-dextrinoid basidiospores and *C.* sect. *Pseudoamyloideae* with species forming dextrinoid basidiospores. In the present study, three species with dextrinoid basidiospores were included in our phylogenetic analyses, namely *C. bucknallii* (the type species of *C.* sect. *Pseudoamyloideae*), *C. icterina*, and *C. rhodella*. However, the three species did not form a clade (Figure 1 and Figure 2) and the sequence of *C. constricta* Singer, the type species of *Cystolepiota*, is not available. Thus, further sampling and phylogenetic analyses are needed to verify and revise the infrageneric classification of *Cystolepiota*.

*Melanophyllum* is distinguished from *Cystolepiota* by the formation of reddish-brown to greenish spore prints and ornamented spores [10,12,17]. In this study, we demonstrate that some species of the *C. seminuda* species complex form ornamented spores as well. Phylogenetic results show that the two genera belong to the same main clade (Figure 2). Specimens previously identified as *Melanophyllum* did not even form a monophyletic clade within *Cystolepiota*, which confirms that the above-mentioned spore characteristics are not genus specific. It seems reasonable to synonymize the two genera. However, the name *Melanophyllum* Velen. 1921 has priority as it was published before *Cystolepiota* Singer 1952, which means that *Cystolepiota* would become a synonym of *Melanophyllum*. It requires careful consideration, if the about 50 current species of *Cystolepiota* should be transferred to *Melanophyllum* or to conserve *Cystolepiota* and subsequently combine the six *Melanophyllum* species in *Cystolepiota*. A more comprehensive and robust phylogeny is necessary before combining these, in order to confirm that whether they are congeneric.

Four accepted species are included in *Melanophyllum* according to Species Fungorum (https://www.speciesfungorum.org/ (accessed on 25 April 2023)). Sequences of two of them, *M. haematospermum* and *M. eyrei*, were included in the phylogenetic analyses of this study. The resulting phylogenies (Figure 1 and Figure 2), indicate that the species diversity of *Melanophyllum* was underestimated and several new species are waiting to be described.

The number of *Cystolepiota* species including *Pulverolepiota* and excluding *Melanophyllum* accepted by different researchers ranges from 10 to 45 [63,64,65,66]. According to Index Fungorum (http://www.indexfungorum.org/ (accessed on 25 April 2023)), 47 *Cystolepiota* species had been described prior to this study, mostly in the last century. Reports of only seven new species have been published in the past 20 years [55,57,58,67,68,69]. The *Cystolepiota* species from East Asia, Europe, India, North America, and South America have currently been well-characterized morphologically [5,12,20,66,67,70,71], while only few species were provided with DNA sequence data. Many species, especially those described from South America, have not been recollected since they were published [20].

Several reasons may lead to this limited knowledge of the species diversity of *Cystolepiota*. (1) The tiny and solitary basidiomata make them easily overlooked in field investigations and mycologists are not inclined to study and describe species based on only one basidioma. For example, *C. oliveirae* was first recorded in China in this paper based on one basidioma with a pileus that was 1.4 cm in diam. Several potential new species belonging to the *C. seminuda* complex, such as KUN-HKAS 56447, KUN-HKAS 84333 and KUN-HKAS 84177, that were included in our ITS phylogeny (Figure 1) were not described in the present paper, because there is only one mature basidioma each available for description. (2) The collections of many species, e.g., *C. seminuda*, *C. hetieri* (Boud.) Singer, and *Melanophyllum haematospermum* (Bull.) Kreisel, apparently exhibit almost identical morphological characteristics but show a large variability in their ITS sequences as demonstrated by the phylogenies of this study (Figure 1). This indicates the existence of many cryptic species. With further investigations and research from less studied regions such as Africa and Australia, more *Cystolepitoa* species are expected to be discovered.

### 4.2. Circumscription of Species in the Cystolepiota seminuda Species Complex

We regard the species in Clade I and Clade II (Figure 2) as the members of the *C. seminuda* species complex that includes three named species: *C. pseudoseminuda*, *C. seminuda*, and a species represented by specimens that had been identified as *Melanophyllum eryei*, as well as ten potential new species according to the sequence similarity. The main characteristics of the specimens belonging to the *C. seminuda* species complex investigated in the present study are: small basidiomata with a whitish pileus covered by pulverulent or granulose squamules that are whitish, cream to pinkish, rarely brownish; free lamellae that are white to cream in most species and greenish in *M. eyrei*; a stipe with a whitish upper portion and a brownish lower portion, which turns purplish or reddish-brown with age or when touched; a stipe often covered with whitish to cream pulverulent squamules; basidiospores which are ellipsoid, smooth-walled, or rough-walled; absent cheilocystidia and pleurocystidia; clamp connections present. However, collections may meet the above descriptions, but still show certain morphological differences, especially regarding the basidiospores.

*Cystolepiota sistrata* has been considered as conspecific with *C. seminuda* [10,15,31,32,33]. We follow the view of Vellinga (1987) [15], who did not use the name *C. sistrata* because the description in the protologue is too general and Fries made the concept of this species even more ambiguous [30,32,34,35]. As with *C. sistrata*, several names have been treated as synonyms of *C. seminuda*, namely: *Lepiota sororia*, *L. seminuda* f. *minima,* and *L. rufipes*. Different treatments of those names and different characteristics used in the circumscriptions of species in the *C. seminuda* complex are discussed below.

The size of basidiomata varies within the *C. seminuda* complex. Some authors consider basidiomata size as an important characteristic for distinguishing *Cystolepiota* species. *Lepiota seminuda* f. *minima* and *Lepiota sistrata* f. *minima* (both Nom. inval., Art. 39.1, Melbourne) are different from *C. seminuda* because of the smaller pileus (8 mm in diam) and the more slender stipe (<1 mm in diam) [72]. However, based on this study, these dimensions are within the morphological variability of *C. seminuda*. In contrast, Huijsman described *Lepiota sororia*, which was considered a synonym of *C. seminuda* by Vellinga [21,33], with a pileus of 12–30 mm diam and a stipe measuring 30–70 × 1–3 mm, which is larger than that determined for *C. seminuda* in this study.

The descriptions of basidiospore size and shape of *C. seminuda* are inconsistent in different studies. Although the spores are basically elliptical, some authors have noticed variations in spore shape [21,66]. Vellinga (1987) described the spores as ellipsoid to oblong and measured 3.5–5.0(–5.5) × 2.0–3.0 µm with a Q value of 1.35–2.1(–2.2); the spores Yang (2019) described were broadly ellipsoid to ellipsoid and measured 3.5–4.5 × 2.5–3 µm, Q value is 1.25–1.67(–1.80). Thus, the specimens studied in the two articles may belong to different species, *C. pseudoseminuda*, *C. seminuda* or further species of the *C. seminuda* species complex. The two species described in this paper, *C. pseudoseminuda* and *C. seminuda*, also vary in size and shape of basidiospores. Based on measurements by Vellinga (1987) [21], the holotype of *Lepiota sororia* has longer and narrower basidiospores: (4.2–)4.6–5.3(–5.4) × (2.1–)2.3–2.9(–3.0) µm, Q = (1.65–)1.7–2.1(–2.2). Vellinga (2010a) also studied the holotype of *Lepiota rufipes* [36], a species described from the USA, which she treated as a synonym of *C. seminuda*. The basidiospores of *L. rufipes* measured 4.1–4.9 × 2.4–2.8 µm, Q = 1.65–2.05, Q_m_ = 1.85, is larger than the basidiospores of *C. seminuda* measured in this study. Some collections regarded as *C. seminuda* formed even larger basidiospores [20,21,73]. Although general differences in shape and size of spores of *C. pseudoseminuda* and *C. seminuda* were determined in this study, considerable variations among different fruit bodies were observed in both species (Figure 6). Thus, when describing new species from the *C. seminuda* complex, as many specimens as possible from different locations need to be examined.

In most previous studies, basidiospores of *C. seminuda* are described as smooth-walled, but in some as smooth-walled or with insignificant warts [12,66]. Based on molecular data, we revealed that the *C. seminuda* complex consists of two clades. Observations of basidiospores with SEM suggest that species of Clade I form ornamented basidiospores, while species of clade II form smooth-walled basidiospores (Figure 1 and Figure 2). Thus, the ornamentation of basidiospores is a key characteristic to distinguish species within the *C. seminuda* complex. In addition, the basidiospores of the type specimen of *L. sororia* have a smooth-walled surface (Figure 4H,I), indicating that this species may belong to clade II.

The basidiospores of all specimens involved in this study are non-dextrinoid. In contrast, basidiospores of the “*C. seminuda*” collections from South America were reported as being dextrinoid [20,73]. This could be an indication that these collections also represent different species. However, whether the chemical reactions of basidiospores are useful for identification of species in this complex requires further study.

According to the discussion above, which was based on the examination of accessible specimens and the literature review, the species in the *Cystolepiota seminuda* complex can be differentiated by a combination of the following characteristics: basidiomata size, spore size and shape, ornamentation of spores, and DNA sequence data. 

The species in the *C. seminuda* complex show different geographic distribution patterns. For example, *C. seminuda* is widely distributed in the temperate and subtropical zones of the Northern Hemisphere, while *C. pseudoseminuda* has so far only been collected in the subtropical zones of China. Further species in this complex are so far only represented by specimens from China and from North America, respectively. This indicates that geographic distribution may support the species differentiation within the *C. seminuda* complex.

## Figures and Tables

**Figure 1 jof-09-00537-f001:**
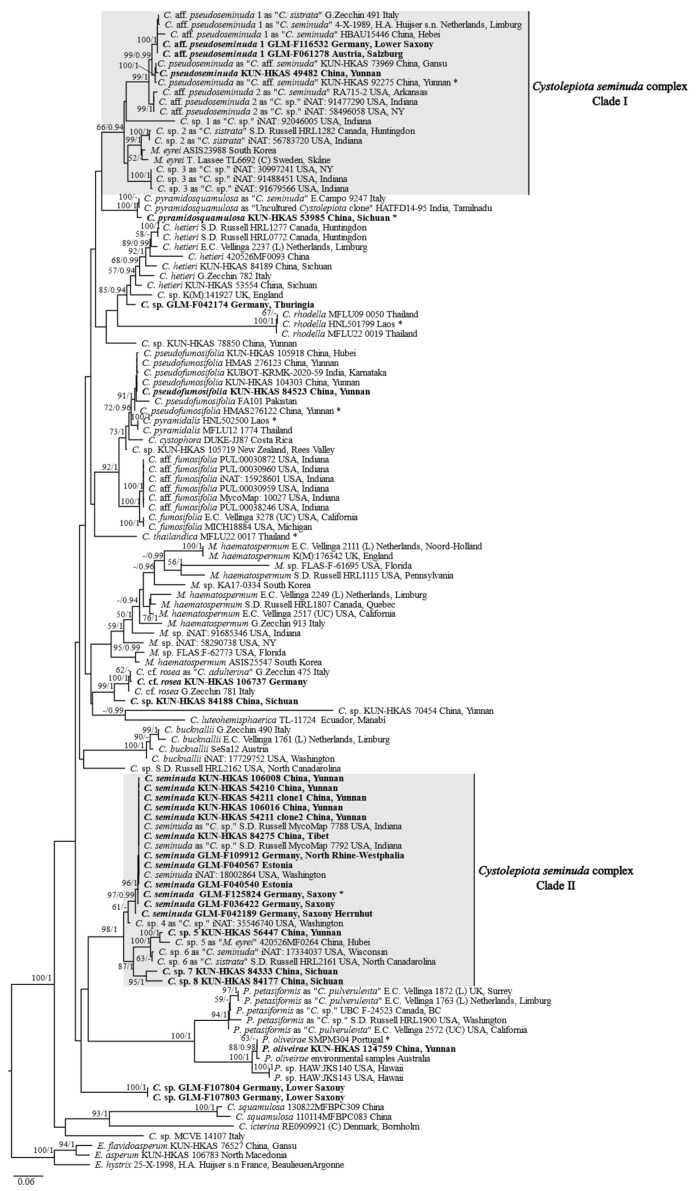
Maximum likelihood tree based on ITS (the nuc rDNA internal transcribed spacer region ITS1-5.8S-ITS2) sequences of specimens of *Cystolepiota* and related genera. Bootstrap values of ML ≥ 50%, and BI ≥ 0.90 are indicated above the branches (ML/BI). Specimens sequenced within this study are in bold. Type collections are marked with an asterisk. The clades representing the *C. seminuda* complex are highlighted by grey boxes. *C*. = *Cystolepiota*; *E*. = *Echinoderma*; *M*. = *Melanophyllum; P*. = *Pulverolepiota*. *E. flavidoasperum*, *E. asperum* and *E. hystrix* are used as outgroup.

**Figure 2 jof-09-00537-f002:**
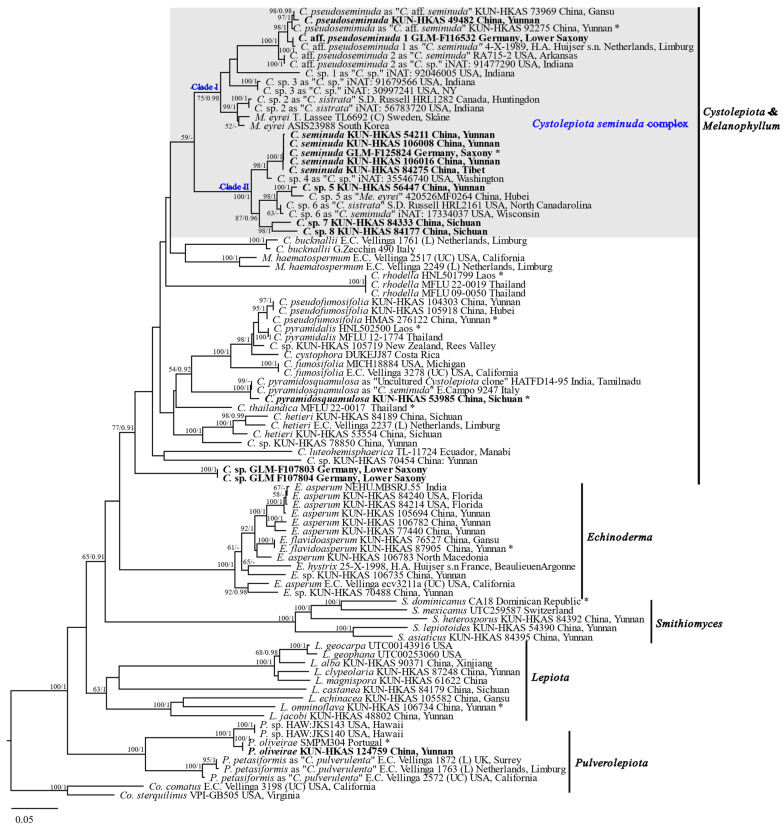
Maximum likelihood tree inferred from a combined data set of ITS, LSU (the D1–D2 domains of nuc 28S rDNA), *rpb2* (the most variable region of the second-largest subunit of RNA polymerase II), and *tef1* (a portion of the translation–elongation factor 1-α) sequences of specimens of *Cystolepiota* and related genera. Bootstrap values of ML ≥ 50%, and BI ≥ 0.90 are indicated above the branches (ML/BI). Specimens sequenced within this study are in bold. Type collections are marked with an asterisk. The clade standing for *C. seminuda* complex is highlighted by a grey box. *C.* = *Cystolepiota*; *Co*. = *Coprinus*; *E*. = *Echinoderma*; *L*. = *Lepiota*; *M*. = *Melanophyllum; P*. = *Pulverolepiota*; *S*. = *Smithiomyces*. *Co. comatus* and *Co. sterquilinus* are used as outgroup.

**Figure 3 jof-09-00537-f003:**
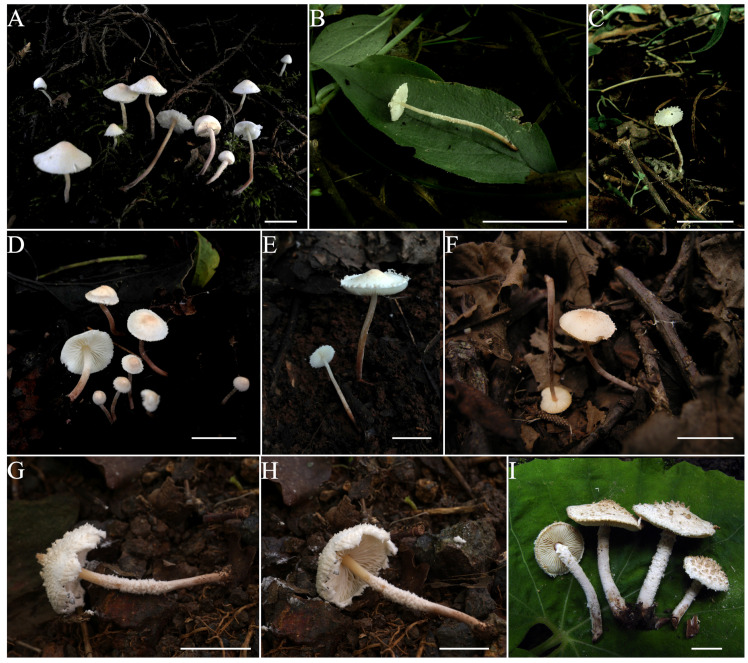
Basidiomata of *Cystolepiota* and *Pulverolepiota* species. (**A**–**C**). *C. pseudoseminuda*. (**A**). KUN-HKAS 92275(Holotype). (**B**,**C**). KUN-HKAS 73969. (**D**–**F**). *C. seminuda*. (**D**). KUN-HKAS 106008. (**E**). KUN-HKAS 106016. (**F**). KUN-HKAS 54211. (**G**,**H**). *P. oliveirae*. KUN-HKAS 124759. (**I**). *C. pyramidosquamulosa*. KUN-HKAS 53985 (Holotype). Bars = 1 cm.

**Figure 4 jof-09-00537-f004:**
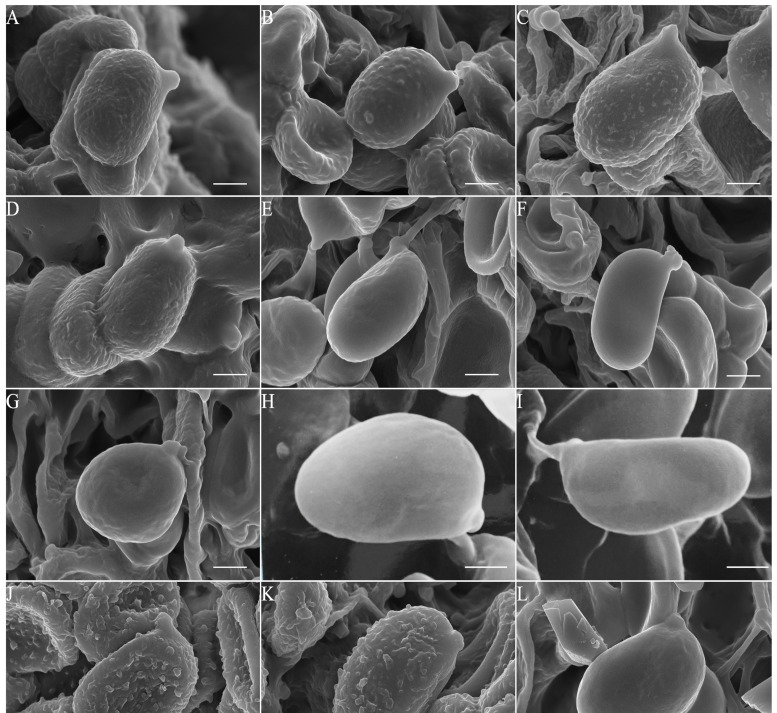
Basidiospores of *Cystolepiota* and *Pulverolepiota* species under SEM. (**A**,**B**). *C.* aff. *pseudoseminuda* 1. (**A**). GLM-F116532. (**B**). GLM-F061278. (**C**,**D**). *C. pseudoseminuda*. (**C**). KUN-HKAS 92275 (Holotype). (**D**). KUN-HKAS 73969. (**E**–**G**). *C. seminuda*. (**E**). KUN-HKAS 106016. (**F**). GLM-F125824 (Neotype). (**G**). GLM-F042189. (**H**,**I**). *Lepiota sororia*. L0054151 (Holotype). (**J**,**K**). *P. oliveirae*. KUN-HKAS 124759. (**L**). *C. pyramidosquamulosa*. KUN-HKAS 53985 (Holotype). Bars = 1 μm.

**Figure 5 jof-09-00537-f005:**
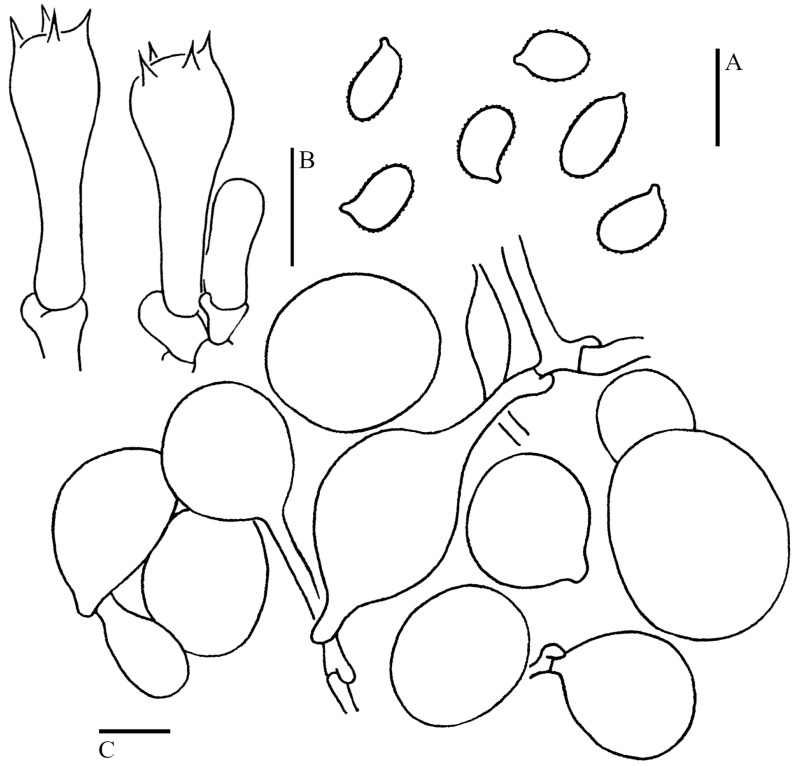
Microscopic features of *Cystolepiota pseudoseminuda* (Holotype, KUN-HKAS 92275). (**A**): Basidiospores. (**B**): Basidia. (**C**): Squamule cells. Bars: (**A**) = 5 μm; (**B**,**C**) = 10 μm.

**Figure 6 jof-09-00537-f006:**
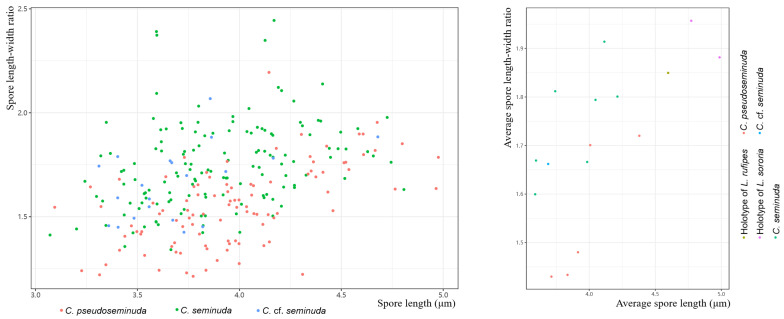
The scatter diagram of spore lengths and spore length–width ratios of species in the *Cystolepiota seminuda* complex.

**Figure 7 jof-09-00537-f007:**
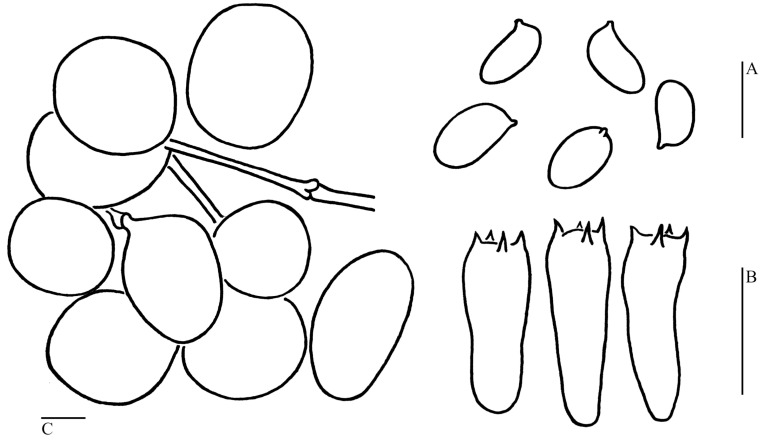
Microscopic features of *Cystolepiota pyramidosquamulosa* (Holotype KUN-HKAS 53985). (**A**): Basidiospores. (**B**): Basidia. (**C**): Squamule cells. Bars: (**A**) = 5 μm; (**B**,**C**) = 10 μm.

**Figure 8 jof-09-00537-f008:**
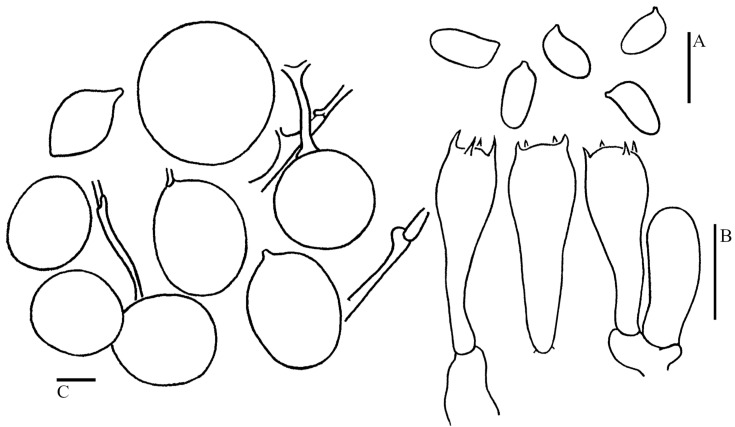
Microscopic features of *Cystolepiota seminuda*. (**A**): Basidiospores. (**B**): Basidia. (**C**): Squamule cells. (**A**,**B**) from GLM-F109912, (**C**) from KUN-HKAS 106016. Bars: (**A**) = 5 μm; (**B**,**C**) = 10 μm.

**Figure 9 jof-09-00537-f009:**
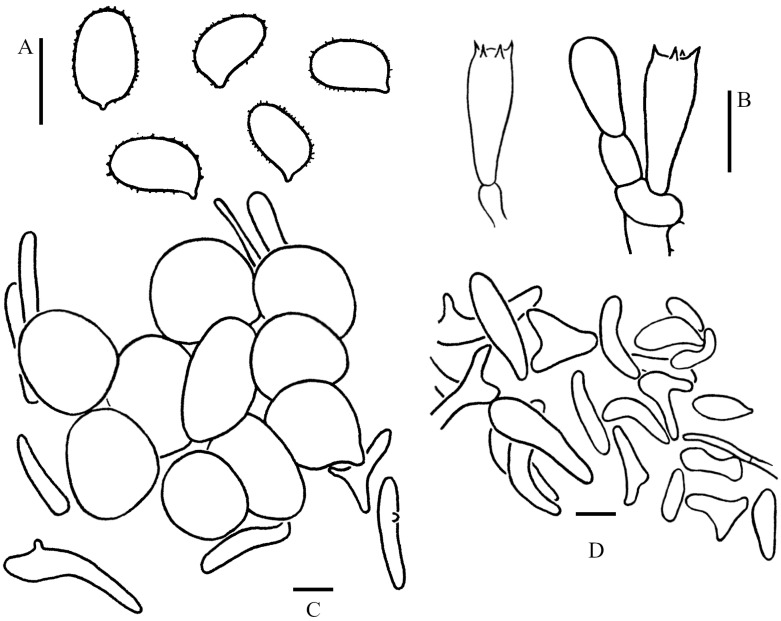
Microscopic features of *Cystolepiota oliveirae* (KUN-HKAS 124759). (**A**): Basidiospores. (**B**): Basidia. (**C**): Tightly arranged squamule cells. (**D**): loosely arranged squamule cells. Bars: (**A**) = 5 μm; (**B**) = 10 μm. (**C**) = 20 μm. (**D**) = 20 μm.

## Data Availability

The sequences presented in this study are openly available in https://www.ncbi.nlm.nih.gov/ (accessed on 25 July 2022) (see the Appendix A for the accession numbers). The alignments and phylogenetic tree files are available in FigShare at https://doi.org/10.6084/m9.figshare.22284127.v1 (accessed on 16 March 2023). All new taxa were registered in MycoBank (http://www.mycobank.org/ (accessed on 16 March 2023)).

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
