# Peer review of "Taxonomy and Phylogeny of Cystolepiota (Agaricaceae, Agaricales): New Species, New Combinations and Notes on the C. seminuda Complex"

_jof, 2023, doi:10.3390/jof9050537_

Round 1
Reviewer 1 Report
The paper presents detailed phylogenetic analyses of Cystolepiota sensu lato, with the description of new species in the Cystolepiota seminuda complex and re-assessment of the status of the genus Pulverolepiota.
Phylogenetic analyses are adequate for the stated purposes of the paper, and taxonomic descriptions and discussions are comprehensive.
I only have a couple of minor suggestions, about some aspects that I think could be discussed a bit more in detail in the paper:
Regarding P. petasiformis – There seems to be a relatively wide variation of ITS sequences in the collections studied here (96-99% similarity) Do the authors think this is one single species? Or do they suspect cryptic taxa might exist in P. petasiformis?
Collections identified as Melanophyllum haematospermum and M. eyrei show a remarkable level of molecular diversity, with more than 10 species-level lineages recovered in the ITS phylogeny. I think some brief comments in the discussion about these taxa would be appreciated. These are considered “easy” to recognize species, and it seems that extensive (morphologically) cryptic diversity exists within these taxa.
Reviewer 2 Report
The manuscript is devoted to the phylogeny of the complex genus Cystolepiota. The taxonomy of this genus has a rich history and requires detailed phylogenetic substantiation. The previously described species do not correspond to the known modern diversity. This condition needs to be improved. The authors have done significant work in this direction. They directed their efforts to intrageneric phylogeny and a detailed study of the most important species complex of Cystolepiota seminuda.
The presentation of the work is clear and concise and allows you to get an idea of both the general design of the study and the detailed features of the proposed species and combinations.
Some of the comments below are not critically important, however may be helpful to better understand the study design.
I believe that the results are largely reproducible based on the research methods described.
However, as can be seen from Appendix Table 2, in the multigene phylogenetic tree, only 39 out of 93 taxa are represented by all four loci.
What allowed the authors to believe that the absence of a significant part of the genetic information did not affect the topology of the combined tree?
In the ITS tree and in the combined tree, I can see some topology differences. How can the authors explain these differences?
It seems to me important to give a method for congruence measuring and show the values obtained.
It would be useful to explain why the authors resorted to a combined tree, instead of two or three trees for individual genes, what advantages the combined tree gave in understanding the phylogeny of the studied group.
It will be useful in additional materials to provide trees for individual loci, in the results to mention the presence and support of large branches in individual trees, and then move on to the combined tree.
Publication of the alignment in accessible repositories will enable readers to gain a better understanding of these new species and their phylogenetic position, and will provide an impetus for future studies of the genus.
The conclusions are consistent with the evidence and arguments presented
Figures, images and schemes are appropriate.
Data acquired from the Genbank is not accompanied by the appropriate references to sources. Primary alignments are not available.
I don't see any ethical issues related to this article.
Remarks:
R. 24 Key words will be “new species” and “new combinations”
R. 83-85. The conclusions made by the authors are broader than the aim. The aim needs to be reformulated according the conclusion.
R. 536.
What are the limits of intraspecific and interspecific variability of sequences in the genus Pulverolepiota? Intraspecific differences between sequences of 4% look too big. Unfortunately, the lack of available alignment does not allow one to get an idea of the length of these sequences and the character of the substitutions. On the ITS tree, this branch does not look homogeneous. Apparently, Pulverolepiota petasiformis is another species complex or the result of significant geographical isolation.
It seems to me useful to discuss this point.
I hope that my suggestions will be useful to the authors.
Best regards.
Reviewer 3 Report
I had the pleasure of reviewing the interesting manuscript of Qu et al., Taxonomy and phylogeny of Cystolepiota (Agaricaceae, Agaricales): new species, new combinations and notes on the C. seminuda complex.
The paper aims to discuss the species Cystolepiota seminuda and its look-alike worldwide, with a focus on two new species from China, and the genus Pulverolepiota, usually synonymized and here shown to represent an autonomous lineage distinct from the former, with two new combinations introduced here.
The article is well-written and nicely illustrated, and the bibliography is rather exhaustive on the subject, including "grey literature", and I personally acknowledge these efforts. The species are described appropriately and nomenclatural rules are applied correctly.
I have only two remarks to address to the authors :
1) In the phylogenetic tree is indicated "Cystolepiota rosea". I presume that it concerns the European taxon Cystolepiota rosea (Rea) Bon, illegitimate (but it could also concern C. rosea Singer, extra-European). The correct name for the European one is Cystolepiota moelleri Knudsen.
2) It is a bit regrettable that the authors did not take the opportunity of sequencing themselves the available types of C. sororia and P. pulverulenta (which is discussed in the text only based on Vellinga's morphological analyses), which might have helped discussing synonymies on updated bases, rather than on now 20-year old morphological revisions. The same for some taxa cited in synonymies (but obviously directly copied from MycoBank or Index Fungorum databases) which might have been available, such as P. roseolanata (type specimens at L) , P. pulverulenta var. subrosea Bon and f. minima Bon, Migl. & Brunori (type specimens at LIP).
3) I don't see clearly the interest of showing a combined phylogram including ITS sequences (fig. 2) when a single ITS phylogeny is already shown in fig. 1. It is assumed that ITS does not provide robust supports on basal branches of the tree, and possibly add more noise than necesary in the combined tree (or, at best does not contribute to its resolution). Fig. 2 might be more robust, and thus a better illustration to the discussion on systematics with LSU, rpb2, and tef1 only (or with 5.8S if really needed but without ITS1/ITS2).
4) The conclusion " It would also be advisable to epitypify the type species of both genera before combining these, in order to confirm that they are congeneric." is not stringent. The designation of epitypes for the type species of Melanophyllum and Cystolepiota will not directly confirm congenericity. Melanophyllum canali (the type species of Melanophyllum) is well-documented enough (as M. haematospermum) to be reasonably considered as stabilized without the necesity of an epitype. Cystolepiota constricta (the type species of Cystolepiota) is quite an enigmatic species without corresponding sequences and its identity as well as the current circumscription of the genus Cystolepiota are still questionnable. But if the authors consider that they bring new elements to this question, it should be mentioned earlier in the discussion, or even in the introduction (nevertheless, if the authors express too many doubts about Cystolepiota as appropriate genus, they must take care of art. 36.1 of the Code of nomenclature which might invalidate their new taxa). It is understandable that teh authors are reluctant to propose such an impacting answer with so partial data, but the real question they should address is if a project of conservation of the name Cystolepiota vs Melanophyllum would be appropriate, otherwise the nomenclatural rules automatically make Melanophyllum the appropriate generic name for the whole clade, whatever the type of Cystolepiota might be. More than epitypifications, a more complete and robust phylogenetic reconstruction would be required for such a systematic update (but this would apply to all Lepiotoid clades in the Agaricaceae, a very ambitious task actually).
In conclusion, I would incitate the authors to take profit of their experience in this clade to test Vellinga's (2006) synonymies by DNA revision of available type collections (and, is possible, reliably identified European and American collections for older taxa, e.g. C. adulterina, C. hetieri etc.) in these clades, what would strenghten and fulfill their taxonomic proposals. I hope they will agree to do that, for a revised version of this paper, of for a very next one which I would warmly support.
Round 2
Reviewer 2 Report
I am satisfied with the answers of the authors and the changes made.
Primary data are now available, references to the authors of the used sequences from the Genbank are given in the supplementary files.
There are two small comments in the attached file.
Best wishes

Author Response
Response to Reviewer 2 Comments
Thank you again for your valuable comments. The manuscript has been revised according to your suggestions. Point-by-point responses are provided below.
Comment 1: I don’t understand whose diversity is referred to in paragraph 3. Maybe swap paragraphs 2 and 3 or add the genus of Cystolepiota in paragraph 3.
Response: We have modified the expression in the revised manuscript.
Comment 2: “C. sp.” should be changed to “Cystolepiota sp.”.
Response: The manuscript has been modified according to the comment.